# Imaging Features of Hepatocellular Carcinoma in the Non-Cirrhotic Liver with Sonazoid-Enhanced Contrast-Enhanced Ultrasound

**DOI:** 10.3390/diagnostics12102272

**Published:** 2022-09-20

**Authors:** Sheng Chen, Yi-Jie Qiu, Dan Zuo, Shuai-Nan Shi, Wen-Ping Wang, Yi Dong

**Affiliations:** Department of Ultrasound, Zhongshan Hospital, Fudan University, Shanghai 200032, China

**Keywords:** hepatocellular carcinoma (HCC), non-cirrhotic liver, contrast-enhanced ultrasound (CEUS), Sonazoid, washout

## Abstract

Purpose: To investigate the Sonazoid-enhanced contrast-enhanced ultrasound (CEUS) features of hepatocellular carcinoma (HCC) in a non-cirrhosis liver background, in comparison to those in liver cirrhosis. Methods: In this retrospective study, 19 patients with surgery and histopathologically proven HCC lesions in non-cirrhosis liver background were included regarding Sonazoid-enhanced CEUS characteristics. Two radiologists evaluated the CEUS features of HCC lesions according to the WFUMB (World Federation of Societies for Ultrasound in Medicine and Biology) guidelines criteria. Thirty-six patients with HCC lesions in liver cirrhosis were included as a control group. Final diagnoses were confirmed by surgery and histopathological results. Results: Liver background of the non-cirrhosis group including normal liver (*n* = 7), liver fibrosis (*n* = 11), and alcoholic liver disease (*n* = 1). The mean size of non-cirrhosis HCC lesions was 60.8 ± 46.8 mm (ranging from 25 to 219 mm). During the arterial phase of Sonazoid-enhanced CEUS, most HCCs in non-cirrhotic liver (94.7%, 18/19) and in cirrhotic liver (83.3%, 30/36) presented non-rim hyperenhancement. During the portal venous phase, HCC lesions in the non-cirrhosis liver group showed relatively early washout (68.4%, 13/19) (*p* = 0.090). Meanwhile, HCC lesions in liver cirrhosis background showed isoenhancement (55.6%, 20/36). All lesions in the non-cirrhotic liver group showed hypoenhancement in the late phase and the Kupffer phase (100%, 19/19). Five cases of HCC lesions in liver cirrhosis showed isoenhancement during the late phase and hypoenhancement during the Kupffer phase (13.9%, 5/36). The rest of the cirrhotic HCC lesions showed hypoenhancement during the late phase and the Kupffer phase (86.1%, 31/36). Additional hypoenhanced lesions were detected in three patients in the non-cirrhosis liver group and eight patients in the liver cirrhosis group (mean size: 13.0 ± 5.6 mm), which were also suspected to be HCC lesions. Conclusions: Heterogeneous hyperenhancement during the arterial phase as well as relatively early washout are characteristic features of HCC in the non-cirrhotic liver on Sonazoid-enhanced CEUS.

## 1. Introduction

Hepatocellular carcinoma (HCC) accounts for the fifth most common tumor worldwide and the second leading cause of cancer-related death. Liver cirrhosis is a key risk factor for HCC and approximately 80–90% of cases of HCC arise in patients with cirrhosis [1]. The reported incidence rate of HCC is 2–4% per year in patients with cirrhosis [2]. However, there are about 20% of HCCs still developing in non-cirrhosis liver background [1]. A variety of etiologic factors can implicate the development of HCC without underlying cirrhosis, such as alcoholic, non-alcoholic fatty liver disease (NAFLD), non-alcoholic steatohepatitis (NASH), autoimmune liver disease, genetic haemochromatosis, alpha-1-antitrypsin deficiency, Wilson’s disease, etc. [3]. Those patients are usually asymptomatic in the early stage due to better hepatic reserve and therefore lacking surveillance [4]. Hence, tumors are generally detected at an advanced stage with larger size. However, the overall survival and disease-free survival of non-cirrhotic patients with HCC are better than those of cirrhotic patients with HCC [5].

According to previous studies, HCC in non-cirrhotic patients usually presents as large hypoattenuating masses that are predominantly solitary or dominant with satellite lesions on unenhanced computed tomography (CT), with being partially encapsulated, and areas of necrosis and hemorrhage [4]. On contrast-enhanced CT, it often shows hyperenhancement during the arterial phase and the portal venous phase, and washout on the delayed phase [4,6,7]. On magnetic resonance imaging (MRI), HCC in a non-cirrhotic liver commonly shows as hypointense on T1-weighted imaging and hypo- or hyperintense on T2-weighted imaging. Other independent features include lack of central tumor enhancement and presence of satellite lesions [8,9]. Compared to HCC in liver cirrhosis, a central scar is more commonly detected in HCC lesions in non-cirrhotic liver on MRI [10]. However, there is no specific diagnostic criteria for HCC in non-cirrhotic liver by CT/MRI.

Contrast-enhanced ultrasound (CEUS) allows the characterization of incidentally detected, indeterminate focal liver lesions (FLLs) with inconclusive findings via CT or MRI [11,12]. Generally, HCC lesions in liver cirrhosis are characterized by arterial phase hyperenhancement, followed by late and mild washout on CEUS [11,13]. Meanwhile, the CEUS features of HCC in non-cirrhotic liver has less been explored.

Sonazoid is a second-generation ultrasound contrast agent, which consists of microbubbles of perfluorobutane gas with phospholipid monolayer shells. Sonazoid microbubbles are characterized by being taken up by Kupffer cells of the liver mononuclear macrophage system. Kupffer-phase imaging is specific to detect HCC lesions as hypoenhancement lesions, which lack Kupffer cells [14]. Stable Kupffer-phase imaging lasts up to 60 min, which is beneficial for a whole-liver scan. It can facilitate detecting indistinctive small or isoechoic HCC lesions on B-mode ultrasound (BMUS), with 73.2–84.6% sensitivity and 95.0–98.6% specificity [15,16,17]. To the best of our knowledge, Sonazoid-enhanced CEUS features of HCC in the non-cirrhotic liver have been rarely reported.

The aim of our study was to investigate the Sonazoid-enhanced CEUS features of histopathologically proven HCC in the non-cirrhotic liver, in comparison to HCC lesions in liver cirrhosis.

## 2. Materials and Methods

### 2.1. Institutional Board Approval

This retrospective study was approved by the institutional review board of our university hospital (ID: B2020-424R). The informed consent was waived. The procedure followed was in accordance with the Declaration of Helsinki.

### 2.2. Patients

From November 2020 to January 2022, patients diagnosed with HCC in our hospital were included and analyzed. The inclusion criteria of this study were as follows: (1) Patients’ age ranged from 18 to 85 years; (2) Conventional BMUS could clearly detect the FLLs; (3) Surgery and histopathological diagnosis could be obtained both of the FLLs and it’s surrounding liver parenchyma; (4) CEUS examination with Sonazoid was performed one week before surgery. 

The exclusion criteria were as follows: (1) Patients have a known history of allergic reaction to perfluorobutane gas or Sonazoid; (2) The lesion was invisible on BMUS; (3) Patients’ lack of final histopathological results.

### 2.3. Ultrasound Examination Technique

Conventional BMUS and CEUS examinations were performed in all patients by an experienced radiologist (more than 20 years of CEUS of the liver), who was aware of the patients’ clinical histories. All ultrasound examinations were performed by two premium ultrasound systems: Acuson Sequoia (Siemens Healthineers, Mountain View, CA, USA; 5C1 convex array probes) and LOGIQ E20 (GE Healthcare, Milwaukee, WI, USA; C1-6 convex array probes). 

CEUS was performed using contrast harmonic real-time imaging at a mechanical index (MI) of 0.20–0.30. Each examination lasted about 20 min after the bolus injection. Sonazoid (perflubutane; GE Healthcare, Oslo, Norway) was used as the contrast agent. For each lesion, a dose of 0.6 mL was injected as a bolus via a 20-gauge intravenous catheter placed in the left cubital vein, and immediately flushed with 5 mL of 0.9% normal saline. Intermittent CEUS images were evaluated during the following 4 phases to characterize the lesion: arterial phase (10–45 s), portal venous phase (45–120 s), late phase (2–5 min) and Kupffer phase (>5 min) [11]. All examinations were digitally recorded for subsequent analysis.

### 2.4. Imaging Analysis

All ultrasound images were evaluated by two independent radiologists, who were blinded to the clinical histories, histopathological results or other imaging findings. On BMUS, the number, location, maximum diameter (mm), echogenicity (homogeneous or heterogeneous; hyperechoic, hypoechoic, or mix-echoic in comparison to the surrounding liver parenchyma), margin (well- or ill-defined appearance) and shape (regular or irregular) were documented. If there were more than one lesion, only the largest one was evaluated. On CEUS, the Sonazoid enhancement patterns and features of FLLs (hyperenhancement, isoenhancement, or hypoenhancement) during various CEUS phases were evaluated according to current WFUMB guidelines [11]. Additionally, the time begin to washout was observed and recorded. Early (<60 s) and late washout were noted.

### 2.5. Statistical Analysis

Continuous parameters were expressed as the mean ± standard deviation (SD). The Student *t*-test was applied to compare the differences between two groups. Not normally distributed data were expressed as median (interquartile range, IQR) and Mann–Whitney U test was used for further analysis. As for categorical parameters, the chi-square test and Fisher exact test were utilized. All statistical analyses were performed with the SPSS software package (version 20.0, IBM, Armonk, NY, USA).

A *p*-value of less than 0.05 was considered statistically significant. 

## 3. Results

### 3.1. Patient Characteristics

From November 2020 to January 2022, 21 patients were enrolled accordingly, among which two cases were excluded since the focal liver lesions could not be clearly detected on BMUS. Finally, 19 patients (15 males and 4 females; mean age: 58.7 ± 14.3, range: 31–82 years) diagnosed with HCC in the non-cirrhotic liver were included. Among them, seven patients had a normal liver and 12 patients had potential disease of non-cirrhotic liver, including liver fibrosis caused by HBV infection (*n* = 11) and alcoholic liver disease (*n* = 1).

Thirty-six patients (26 males and 10 females; mean age: 57.6 ± 9.2, range: 28–73 years) with histopathologically proven HCC lesions in liver cirrhosis were included as the control group. 

Baseline characteristics of patients including gender, age, alpha-fetoprotein (AFP), carcinoembryonic antigen (CEA), carbohydrate antigen 19-9 (CA19-9) as well as histopathological results are shown in Table 1.

### 3.2. Histopathological Findings

All lesions were confirmed to be HCC histopathologically. In the non-cirrhosis group, the HCC tumor grading included HCC grade II (*n* = 15) and grade III (*n* = 4). The finally confirmed fibrotic stages of non-tumoral liver parenchyma included fibrosis S0 (*n* = 8), S1 (*n* = 4) and S3 (*n* = 7). In the cirrhosis group, the HCC tumor grading included HCC grade I (*n* = 1), grade II (*n* = 28) and grade III (*n* = 7).

### 3.3. BMUS Features

In both groups, most patients had a single lesion (84.2%, 16/19), and most lesions were located in the right lobe of the liver (78.9%, 15/19). The mean size of non-cirrhosis HCC lesions was 60.8 ± 46.8 mm (ranging from 25 to 219 mm). Most of the lesions were homogeneous and predominantly hypoechoic with ill-defined margins and irregular shapes. BMUS findings are detailed in Table 2.

### 3.4. CEUS Features

After injection of Sonazoid, during the arterial phase of CEUS, most of the HCC lesions in the non-cirrhotic liver (94.7%, 18/19) and in the cirrhotic liver (83.3%, 30/36) displayed non-rim hyperenhancement compared to the surrounding liver parenchyma.

During the portal venous phase, most of the HCC lesions in the non-cirrhotic liver (68.4%, 13/19) showed relatively early washout and became hypoenhanced (Figure 1), while most HCC lesions in liver cirrhosis (55.6%, 20/36) displayed isoenhancement (*p* = 0.090) (Figure 2).

During the late phase and the Kupffer phase, all HCC lesions in the non-cirrhotic liver (100%, 19/19) showed hypoenhancement. Five cases of HCC lesions in liver cirrhosis showed isoenhancement in the late phase (13.9%, 5/36) and hypoenhancement in the Kupffer phase. The rest of the cirrhotic HCC lesions showed hypoenhancement during the late phase and the Kupffer phase (86.1%, 31/36) (Table 3). In the Kupffer phase, 15 additional hypoenhanced FLLs were detected in three patients in the non-cirrhosis liver group and in eight patients in the liver cirrhosis group, which were also suspected to be HCC lesions. The mean size of these lesions was 13.0 ± 5.6 mm (ranging from 5 to 25 mm). The mean depth to the liver capsule was 22.3 ± 3.9 mm (ranging from 15 to 28 mm). All the additional lesions showed as isoechoic (100%, 15/15) on BMUS.

### 3.5. Time Begin to Washout

The mean time begin to washout of HCC in non-cirrhosis liver background were 66.5 ± 36.3 s (ranging from 23 to 177 s), which were earlier than the control group (108.6 ± 89.3 s) (ranging from 27 to 467 s) (*p* = 0.012). In the non-cirrhotic liver group, early washout (<60 s) showed no obvious correlation to the liver parenchyma fibrosis stage or HCC tumors’ grades (Table 4).

## 4. Discussion

According to the current American Association for the Study of Liver Disease (AASLD) guidelines, for all chronic hepatitis B patients without cirrhosis, HCC surveillance is recommended for Asian men older than 40 years, Asian women older than 50 years, and Africans [2]. With the progress of liver imaging techniques, the accurate detection of HCC at an early stage has been promoted [18,19].

In recent years, depending on the analysis of CEUS features and CEUS liver imaging reporting and data system (LI-RADS) criteria, the diagnosis of HCC in liver cirrhosis could be made noninvasively before operation, with 85% sensitivity and 91% specificity [11,20,21]. While comparing CEUS features between the non-cirrhosis liver and cirrhosis liver groups, arterial phase hyperenhancement (APHE) could be generally observed in all patients [11,22]. The arterial phase appearances of HCC in non-cirrhotic patients are similar to those in cirrhotic liver, mostly showing rapid and heterogeneous hyperenhancement. Hepatocellular nodules progress in five stages during hepatocarcinogenesis: regenerative nodule, low-grade dysplastic nodule (DN), high-grade DN, early HCC, and progress HCC [23]. Based on histopathological progress, regenerative nodules and DNs derive a major blood supply from the portal vein, while HCC derives a higher proportion of blood supply from the hepatic artery. Formation of many unpaired arteries through neoangiogenesis is a key feature of progression from DN to HCC, which results in APHE of HCC [24].

In non-cirrhotic patients, the washout of HCC tended to start earlier, usually before 60 s. For HCC lesions in liver cirrhosis, the mild washout was mostly observed in the late portal venous phase or late phase, which was usually later than 60 s. Compared with HCC patients in the cirrhosis group, HCC in the non-cirrhosis liver group was characterized by relatively early washout. Compared with the background liver, HCC receives less contrast-enhanced blood during the venous phase due to a relatively diminished portal venous supply and shows hypoenhancement [24]. In the non-cirrhosis liver group, the time to achieve peak enhancement may be earlier than for those in the cirrhotic liver group, resulting in the relatively early washout.

According to current guidelines, the washout feature of CEUS is an important clue for distinguishing malignant from benign FLLs [11,25]. However, previous studies also showed that not all HCCs show washout during the late phase. Washout is observed overall in approximately half the cases of HCC, and rarely in small nodules [26,27]. It has also been reported that well-differentiated HCCs tend to show isoenhancement in the late phase [11]. In our results, five cases of HCC lesions in the liver cirrhosis group were isoenhancement in the late phase and showed washout during the Kupffer phase. The Kupffer phase of Sonazoid-enhanced CEUS allows the whole-liver scan after 2 min; washout could be observed in some isoenhanced HCC lesions during the late phase. Comparing to pure blood-pool ultrasound contrast agents, the unique Kupffer phase of Sonazoid-enhanced CEUS may potentially allow the assessment of washout features for a longer period of time [28]. According to previous reports, nearly one-third of HCCs in liver cirrhosis show washout not in the late vascular phase but in the Kupffer phase [14,28,29]. Therefore, Kupffer-phase imaging facilitates observing late washout and making diagnosis of HCC.

In our patients, additional hypoenhanced FLLs were detected in three patients in the non-cirrhosis liver group and in nine patients in the liver cirrhosis group during the Kupffer phase, which were also suspected to be HCC lesions. On further analysis, the reasons why these lesions were invisible on BMUS included small size (mean size: 13.0 ± 5.6 mm), close to the liver capsule (mean depth: 22.3 ± 3.9 mm), isoechoic (100%, 15/15), etc. With Kupffer-phase imaging, whole-liver scanning helps to detect those small and invisible HCC nodules, since these lesions appear as hypoenhanced lesions with a clear margin [11]. Sonazoid-enhanced CEUS was reported to significantly improve the number of lesions detected during the Kupffer-phase whole-liver scanning by 14% [15]. Hence, it is helpful in detecting additional small or invisible HCC lesions which could be hard to detect on the BMUS scan. 

There are several limitations in our study. First of all, this is a single-center study with a relatively small sample size. Further large-scale multi-center clinical study will be necessary to validate the CEUS diagnostic criteria in HCC in non-cirrhotic livers. Secondly, the possibility of a selection bias cannot be avoided in our retrospective study.

In conclusion, in comparison with HCC in liver cirrhosis, heterogeneous hyperenhancement during the arterial phase as well as relatively early washout during the portal venous phase are characteristic features of HCC in the non-cirrhotic liver. Kupffer-phase whole-liver scanning is helpful for observing late washout and detecting small or invisible HCC lesions on BMUS. 

## Figures and Tables

**Figure 1 diagnostics-12-02272-f001:**
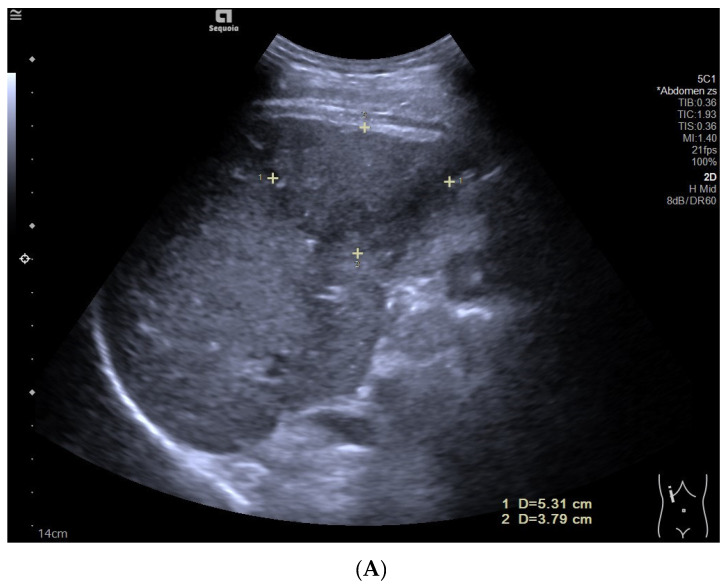
Hepatocellular carcinoma in the non-cirrhotic liver in a 65-year-old male patient. B-mode ultrasound (BMUS) displayed a hyperechoic focal liver lesion in the right lobe of the non-cirrhotic liver, with an ill-defined margin and irregular shape (**A**). Dotted color flow signals could be detected inside the lesion (**B**). On Sonazoid-enhanced contrast-enhanced ultrasound (CEUS), the lesion showed heterogeneous hyperenhancement during the arterial phase (**C**) and relatively early washout (23 s after the injection of Sonazoid). The lesion showed hypoenhancement during the portal venous phase (**D**), late phase (**E**), and Kupffer phase (**F**).

**Figure 2 diagnostics-12-02272-f002:**
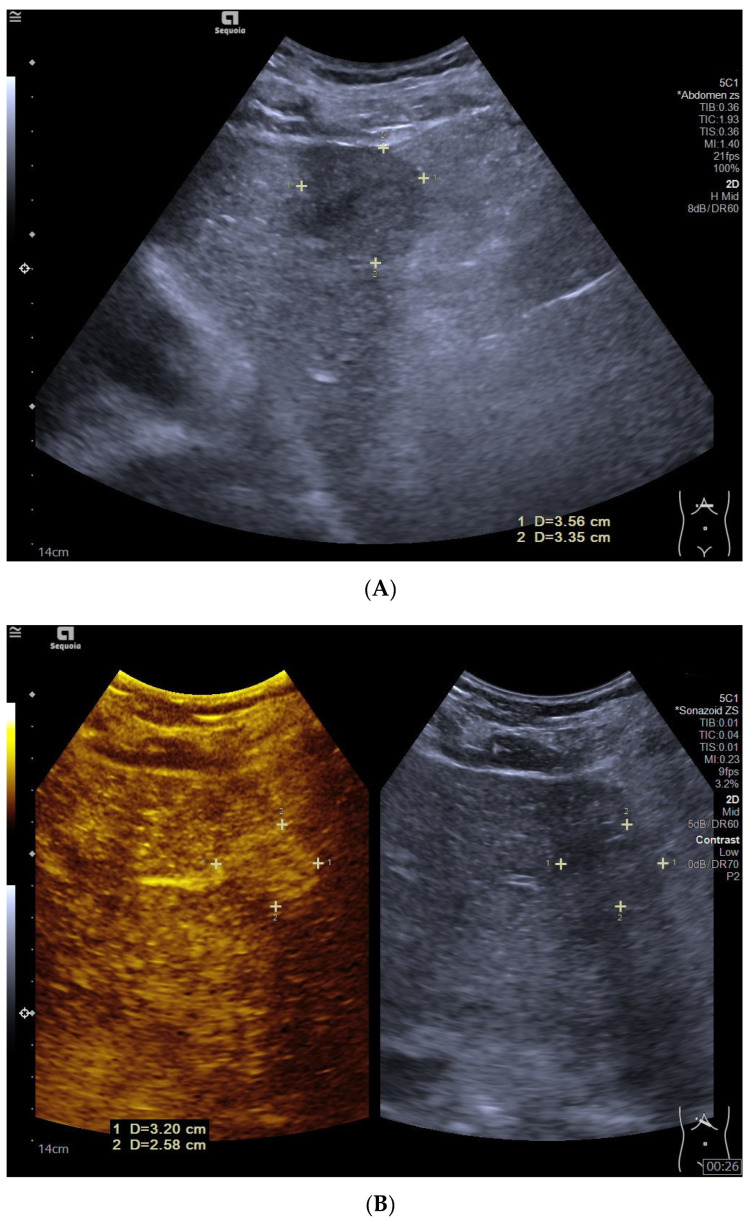
Hepatocellular carcinoma in liver cirrhosis in a 58-year-old male patient. B-mode ultrasound (BMUS) showed a hypoechoic lesion with an ill-defined margin and irregular shape in the left lobe of the liver with cirrhosis (**A**). After injection of Sonazoid, the lesion displayed heterogeneous hyperenhancement during the arterial phase (**B**) and isoenhancement during the portal venous phase (**C**) and late phase (**D**). The lesion showed late washout (264 s after the injection of Sonazoid) and hypoenhancement in the Kupffer phase (**E**).

**Table 1 diagnostics-12-02272-t001:** Base characteristics of patients.

Characteristics	HCC in the Non-Cirrhotic Liver (*n* = 19 Patients)	HCC in the Cirrhotic Liver (*n* = 36 Patients)	*p*-Value
Gender			0.827
Male	15 (78.9%)	26 (72.2%)	
Female	4 (21.1%)	10 (27.8%)	
Age (year)			0.764
Mean ± SD	58.7 ± 14.3	57.6 ± 9.2	
Range	31–82	28–73	
AFP			0.055
Normal	12 (63.2%)	13 (36.1%)	
Elevated	7 (36.8%)	23 (63.9%)	
CEA			0.068
Normal	16 (84.2%)	36 (100%)	
Elevated	3 (15.8%)	0	
CA19-9			0.641
Normal	17 (89.5%)	29 (80.6%)	
Elevated	2 (10.5%)	7 (19.4%)	
Final diagnosis			0.897
Surgery	17 (89.5%)	34 (94.4%)	
Core needle biopsy	2 (10.5%)	2 (5.6%)	

HCC, hepatocellular carcinoma; AFP, alpha fetoprotein; CEA, carcinoembryonic antigen; CA19-9, carbohydrate antigen 19-9.

**Table 2 diagnostics-12-02272-t002:** BMUS features between two groups.

BMUS Features	HCC in the Non-Cirrhotic Liver (*n* = 19 Patients)	HCC in the Cirrhotic Liver (*n* = 36 Patients)	*p*-Value
Number			0.165
Single	16 (84.2%)	24 (66.7%)	
Multiple	3 (15.8%)	12 (33.3%)	
Location			1.000
Left lobe	4 (21.1%)	9 (25.0%)	
Right lobe	15 (78.9%)	27 (75.0%)	
Diameter (mm)			0.075
Mean	60.8 ± 46.8	42.5 ± 27.8	
Range	25–219	10–140	
Echogenicity			0.918
Hyperechoic	5 (26.3%)	9 (25.0%)	
Hypoechoic	11 (57.9%)	23 (63.9%)	
Mix-echoic	3 (15.8%)	4 (11.1%)	

BMUS, B-mode ultrasound; HCC, hepatocellular carcinoma.

**Table 3 diagnostics-12-02272-t003:** CEUS features between two groups.

CEUS Features	HCC in the Non-Cirrhotic Liver (*n* = 19 Patients)	HCC in the Cirrhotic Liver (*n* = 36 Patients)	*p*-Value
Arterial phase			0.696
Hyperenhancement	19 (100%)	33 (91.7%)	
Isoenhancement	0	2 (5.6%)	
Hypoenhancement	0	1 (2.8%)	
Portal venous phase			0.090
Isoenhancement	6 (31.6%)	20 (55.6%)	
Hypoenhancement	13 (68.4%)	16 (44.4%)	
Late phase			0.226
Isoenhancement	0	5 (13.9%)	
Hypoenhancement	19 (100%)	31 (86.1%)	
Kupffer phase			/
Hypoenhancement	19 (100%)	36 (100%)	

CEUS, contrast-enhanced ultrasound; HCC, hepatocellular carcinoma.

**Table 4 diagnostics-12-02272-t004:** Rate of early washout according to liver fibrosis stage and HCC tumors’ histopathologic grades in the non-cirrhosis group.

CEUS Washout Feature	Liver Fibrosis Stage (*n* = 19)			Edmondson Grade of HCC (*n* = 19)	
	S0 (*n* = 8)	S1 (*n* = 4)	S3 (*n* = 7)	Grade II (*n* = 15)	Grade III (*n* = 4)
Washout time					
Early washout <60 s	4 (50.0%)	1 (25.0%)	3 (42.9%)	8 (53.3%)	3 (75.0%)
Late washout >60 s	4 (50.0%)	3 (75.0%)	4 (57.1%)	7 (46.7%)	1 (25.0%)
Washout intensity					
Mild	3 (37.5%)	1 (25.0%)	0	3 (20.0%)	1 (25.0 %)
Marked	5 (62.5%)	3 (75.0%)	7 (100%)	12 (80.0%)	3 (75.0%)

HCC, hepatocellular carcinoma; CEUS, contrast-enhanced ultrasound.

## Data Availability

Data available on request due to restrictions ethical. The data presented in this study are available on request from the corresponding author. The data are not publicly available due to restrictions ethical.

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
