# Peer review of "Imaging Features of Hepatocellular Carcinoma in the Non-Cirrhotic Liver with Sonazoid-Enhanced Contrast-Enhanced Ultrasound"

_diagnostics, 2022, doi:10.3390/diagnostics12102272_

Round 1

Reviewer 1 Report

Dear Authors 

congratulations to your study - i believe this is an important step you made

I do agree with your findings and congratulate you for explaining the features of CEUS in this specific situations for the reader.

I have some small comments to your study and maybe you could add some information to make your results even better

1. Inclusion > you obvously included those patients in the time frame where the lesions were detectable in B-mode ultrasound, which makes sense > how many patients did you see in the same time frame where the lesions could not be detected and therefore CEUS was not possible?

2. Confirmation on histology. All lesions were biopsied or operated - as far as I understand this means target lesions. In some patients you described detection of more lesions on CEUS > were those all correlated intraop or by biopsy? I am asking because my group has done a study on CRLM and intraop CEUS and we found 30% more lesions than on top quality imaging - maybe you can comment on that and add this to the discussion

3. Kupffer Phase > maybe you can add one aspect to the discussion - as the kupffer phase has been very sensitive in your study - what about the difference of sonazoid to sonovue (this is the contrast used in the rest of the world outside Asia, not allowed elsewhere) - I believe this would make an important point in terms of comparing reslults

4. the US guidelines are usually used by the gastroenterologists to state that a biopsy of an HCC is mandatory if there is no signs of cirrhosis before surgery. What is your view on that and can CEUS provide info that makes biopsy unnecessary potentially?

Again, congratulations to this study. Let me just add that I would recommend to use not only sensitivity and specicifity of imaging studies but rather Accuracy as the best comparable parameter in your future research.

Thank you

Author Response

Dear Reviewer,

Thank you for the comments concerning our manuscript entitled " Imaging features of hepatocellular carcinoma in the non-cirrhotic liver with Sonazoid enhanced contrast enhanced ultrasound " (Manuscript ID diagnostics-1862853). The comments have been important and helpful to improve our paper. Please find our point-to-point reply below.

Thank you and best regards!

Prof. Dr. med. Yi Dong

Point 1: Inclusion > you obviously included those patients in the time frame where the lesions were detectable in B-mode ultrasound, which makes sense > how many patients did you see in the same time frame where the lesions could not be detected and therefore CEUS was not possible?

Response 1: From November 2020 to January 2022, 21 patients were enrolled accordingly, among which 2 cases were excluded since the focal liver lesions could not be clearly detected on B mode ultrasound. We made changes in our revised manuscript accordingly.

Point 2: Confirmation on histology. All lesions were biopsied or operated - as far as I understand this means target lesions. In some patients you described detection of more lesions on CEUS > were those all correlated intraop or by biopsy? I am asking because my group has done a study on CRLM and intraop CEUS and we found 30% more lesions than on top quality imaging - maybe you can comment on that and add this to the discussion.

Response 2: Since most of focal liver lesions included in our study were HCC, which were commonly single lesion. For those lesions detected during Kupffer phase, all lesions were confirmed by surgery and histopathological results. We added comments accordingly in our revised manuscript.

Point 3: Kupffer Phase > maybe you can add one aspect to the discussion - as the kupffer phase has been very sensitive in your study - what about the difference of sonazoid to sonovue (this is the contrast used in the rest of the world outside Asia, not allowed elsewhere) - I believe this would make an important point in terms of comparing results.

Response 3: According to a review by Barr et el, when compared to pure blood pool contrast ultrasound agents, the unique Kuppfer phase of Sonazoid enhanced CEUS allow the assessment of washout features for a longer time. We added comments in our revised manuscript accordingly.

Point 4: the US guidelines are usually used by the gastroenterologists to state that a biopsy of an HCC is mandatory if there is no signs of cirrhosis before surgery. What is your view on that and can CEUS provide info that makes biopsy unnecessary potentially?

Response 4: According to current WFUMB guidelines, CEUS is recommended as the first-line imaging method for the characterization of incidentally detected, indeterminate focal liver lesions in patients with a non-cirrhotic liver and without a history or clinical suspicion of malignancy. CUES is proved to be helpful for detecting additional lesions and excluding such benign FLLs with typical CEUS features, such as hemangioma, focal nodular hyperplasia, and hepatocellular adenoma.

Point 5: Let me just add that I would recommend to use not only sensitivity and specificity of imaging studies but rather Accuracy as the best comparable parameter in your future research.

Response 5: Thank you for helpful suggestion. Accuracy is a very important parameter in imaging studies. However, our current study is a pilot study with limited cases. In our future study, we will improve based on your suggestions.

Reviewer 2 Report

Sonazoid is a second-generation ultrasound contrast agent that usually enhances results obtained by  CT, so it could increase the quality of results using CT. I think that such examples presented in this manuscript should be add to clinical equipment (CT), and this is not a biological marker important in disease development. Please Editors decide if this manuscript should be published. 

Author Response

Dear Reviewer,

Thank you for the comments concerning our manuscript entitled " Imaging features of hepatocellular carcinoma in the non-cirrhotic liver with Sonazoid enhanced contrast enhanced ultrasound " (Manuscript ID diagnostics-1862853). The comments have been important and helpful to improve our paper. Please find our point-to-point reply below.

Thank you and best regards!

Prof. Dr. med. Yi Dong

Point 1: Sonazoid is a second-generation ultrasound contrast agent that usually enhances results obtained by CT, so it could increase the quality of results using CT. I think that such examples presented in this manuscript should be add to clinical equipment (CT), and this is not a biological marker important in disease development. Please Editors decide if this manuscript should be published.

Response 1: According to current WFUMB guidelines, the result of Sonazoid enhanced CEUS is not related to CT scan results. We aimed to investigate the Sonazoid enhanced CEUS features of histopathologically proven HCC in the non-cirrhotic liver, in comparison to HCC lesions in liver cirrhosis.

Reviewer 3 Report

thank you for your interesting study. 

points to consider

1) do the authors in this series use any other CEUS agent besides sonozoid for their CEUS? Is this a routine in their clinical practice?

2) There are a few patients who had core needle biopsy for confirmation of HCC diagnosis (table 1). Can the authors comment if there were core biopsies also performed on the non tumour parenchyma at the same time as usually these core biopsies would not include significant amounts of non tumoral parenchyma for histological confirmation of the surrounding parenchyma and may interfere with results.

3) did the authors consider matching the cirrhotic and non cirrhotic groups to control for the grade of tumours and its differentiation and perhaps in 1:2 or 1:3 ratio? this may allow for a more robust conclusion to be made especially in a study with limited numbers of cases in the non cirrhotic group. 

4) typo for page 2 line 65 "benn", line 71 "benifit'. 

Author Response

Dear Reviewer,

Thank you for the comments concerning our manuscript entitled " Imaging features of hepatocellular carcinoma in the non-cirrhotic liver with Sonazoid enhanced contrast enhanced ultrasound " (Manuscript ID diagnostics-1862853). The comments have been important and helpful to improve our paper. Please find our point-to-point reply below.

Thank you and best regards!

Prof. Dr. med. Yi Dong

Point 1: do the authors in this series use any other CEUS agent besides sonozoid for their CEUS? Is this a routine in their clinical practice?

Response 1: In our study, we did not use other contrast agents. According to IRB of our institution, different kinds of contrast agents are not allowed to use in the same patient or in the same time in clinical practice.

Point 2: There are a few patients who had core needle biopsy for confirmation of HCC diagnosis (table 1). Can the authors comment if there were core biopsies also performed on the non tumour parenchyma at the same time as usually these core biopsies would not include significant amounts of non tumoral parenchyma for histological confirmation of the surrounding parenchyma and may interfere with results.

Response 2: Among patients with core needle biopsy and histopathologically confirmed HCC diagnosis, the histopathological analyses were performed on both tumor tissue and non-tumoral parenchyma. Tumor tissue was obtained for the diagnosis of focal liver lesions, and non-tumoral parenchyma was acquired for the confirmation of liver cirrhosis or not.

Point 3: did the authors consider matching the cirrhotic and non cirrhotic groups to control for the grade of tumours and its differentiation and perhaps in 1:2 or 1:3 ratio? this may allow for a more robust conclusion to be made especially in a study with limited numbers of cases in the non cirrhotic group.

Response 3: Thank you for kind comments. Since HCC in the non-cirrhotic liver is relatively rare in clinical practice, the sample size in our current study is limited. In the future study, we will try our best to validate the results based on a large-scale multi-center clinical study.

Point 4: typo for page 2 line 65 "benn", line 71 "benifit".

Response 4: Sorry for these mistakes. We have corrected according to your suggestion.